# The Effect of Land-Use Categories on Traffic Noise Annoyance

**DOI:** 10.3390/ijerph192315444

**Published:** 2022-11-22

**Authors:** Christoph Lechner, Christian Kirisits

**Affiliations:** 1LMU University Hospital Munich, Institute and Outpatient Clinic for Occupational, Social and Environmental Medicine, 80336 Munich, Germany; 2Office of the Tyrolean Regional Government, Department for Emission, Safety and Sites, 6020 Innsbruck, Austria; 3Kirisits Consulting Engineers, 1030 Vienna, Austria; 4Department of Radiation Oncology, Medical University of Vienna, 1090 Vienna, Austria

**Keywords:** land-use planning, spatial planning, noise annoyance, traffic noise

## Abstract

Land-use categories are often used to define the exposure limits of national environmental noise policies. Often different guideline values for noise are applied for purely residential areas versus residential areas with mixed-use. Mixed-use includes living plus limited activities through crafts, commerce, trade, agriculture, and forestry activities. This differentiation especially when rating noise from road, railway, and air traffic might be argued by different expectations and therefore noise annoyance in those two categories while scientific evidence is missing. It should be tested on empirically derived data. Surveys from two studies in the state of Tyrol in urban and rural areas were retrospectively matched with spatial data to analyze the potential different influences on noise effects. Using non-parametric tests, the correlation between land-use category on self-reported noise sensitivity and noise annoyance was investigated. Exposure–response for the two analyzed land-use categories showed no significant impact on noise sensitivity and exposure–response relationships for the three traffic noise sources. Including only noise annoyance, there is not sufficient evidence to define different noise policies for those two land-use categories.

## 1. Introduction

Noise and its effects are a public health issue [1]. The WHO has described health effects in their environmental noise guidelines for the European region [2]. Evidence exists for the most harmful effects of annoyance, sleep disturbance, and ischemic heart disease [2,3,4,5,6]. In terms of the number of people affected, annoyance and sleep disturbances cause by far the most DALYs [7]. Environmental noise research often presents the degree of noise annoyance as exposure–response curves (ERC). The probability of being highly annoyed is estimated as a function of a noise index. A meta-analysis by Guski et. al. [5] provides a systematic overview of environmental noise and annoyance which has been the basis for these recent WHO guidelines. In addition to noise exposure, there are several influencing factors that affect the degree of annoyance. Fields showed that noise annoyance is related to the amount of isolation from sound at home and to five attitudes (fear of danger from the noise source, noise prevention beliefs, general noise sensitivity, beliefs about the importance of the noise source, and annoyance with non-noise impacts of the noise source). Annoyance is not affected to an important extent by ambient noise levels, the amount of time residents are at home, the type of interviewing method, or any of the nine demographic variables (age, sex, social status, income, education, home ownership, type of dwelling, length of residence, or receipt of benefits from the noise source) [8].

One important determinant for noise annoyance is noise sensitivity. The role of noise sensitivity in the noise–response relation was investigated by a comparison of three international airport studies [9]. It seems that noise sensitivity influences noise annoyance irrespective of noise level: at all levels of noise intensity, noise-sensitive people are likely to be more annoyed than the general population. The results of survey data of persons living in the vicinity of Frankfurt Airport Study indicate that noise sensitivity is a reliable predictor of responses to noise [10]. In recent studies in Tyrol, sensitivity to noise was detected as a significant determinant for noise annoyance to all traffic sources in both settings [11,12].

Socioeconomic status (SES) can also be a determinant of exposure. At the same time, SES is influencing both individual susceptibility and resilience [13].

Kroesen et al. developed a causal model for aircraft noise annoyance. Here, perceived disturbance as well as perceived control and coping capacity affect noise annoyance. For perceived disturbance, noise sensitivity and noise exposure contribute. The perceived control and coping capacity depend on attitudes toward the source, authorities, and noise policies, as well as other non-acoustic factors [14]. Riedel et al. defined ‘non-acoustic factors’ as all factors other than the objective, measured, or modeled acoustic parameters which influence the process of perceiving, experiencing, and/or understanding an acoustic environment in context, without being part of the causal chain of this process. This means that ‘non-acoustic factors’ add to or alter the strength or even the direction of effect of the acoustic parameters on a selected health outcome. This moderation effect is linked to the higher risk of adverse health response that features ‘vulnerability’. Ideally, ‘non-acoustic factors’ are not associated with the exposure and are, therefore, not only another (preliminary) response to the exposure [15]. In this context, a land-use category is not a ‘non-acoustic factor’ according to this literature. Noise sensitivity and attitudes toward the noise source are personal factors.

In many countries, noise regulations are formulated based on different parameters, mostly in the form of planning, guideline, and limit values. Such values serve as the basis for environmental impact analysis of new infrastructure or the need for measures to reduce noise from existing sources. Land-use categories often form the basis for such guideline values for noise.

The land-use category by itself is just an attribute in a plan. Certain uses are allowed within the categorized area but not necessarily associated with it. Often a purely residential area is associated with a certain building structure and more green space. These factors may or may not apply. So, it is important to distinguish between the defined land use and the actual situation on-site. Noise levels associated with urban land use were investigated in different areas in Halifax [16]. In residential areas, the noise exposure limits, which were adopted from the Italian legislation were exceeded in the day- and night-time as in mixed-use areas as well. This study provides important evidence concerning the relationship between land use and environmental noise. A planning strategy focused on mixed-use development may result in an increase in noise levels and human exposure to noise at levels with potential health implications.

The data set of the Swiss SIRENE study on road traffic, railway, and aircraft noise annoyance [17] were completed with a wide range of “green” metrics, that includes various metrics for residential green or blue, i.e., water bodies, and their association with annoyance was investigated [18]. These “green” metrics were strongly linked to annoyance. For road traffic noise, visible vegetation and accessibility of green spaces particularly strongly reduce annoyance in cities. Quiet green spaces seem to be more effective in rural areas [5].

An I-INCE report gives results of a survey of legislation, regulations, and guidelines for the control of community noise [19]. There are varying metrics used for the exposure. Noise indicators such as L_den_ and L_night_ are widely used while L_Aeq,24_ is an often used alternative. The range of the limit values during daytime or L_den_ scores from 40 dB to 72 dB in areas with residential use and vary for different sources, whereas night noise limits are regularly 10 dB lower. Many countries define noise level limits depending on categories of land-use zones. Zones with residential use only are distinguished from zones from areas with some additional retail and light-manufacturing businesses. They can also be described as mixed residential zones. The difference between the limit values is mainly 5 dB. Obviously, a lower noise sensitivity of the resident is assumed, or higher impacts are permitted because of the higher activities and ambient noise. Some countries take background sound levels into account for industrial noise assessment. The recent recommendations of WHO [2] are not differentiated based on land use.

There is some evidence that noise may be differentially distributed across communities based on socioeconomic status [20]. Ambient noise levels may also be associated with land-use categories. A land-use regression (LUR) model is applied in estimating environmental exposures to noise [21]. The spatial variability of environmental noise levels was modeled in Montreal, Canada, using noise measurements and land-use characteristics [22]. The main predictors of measured noise levels were road traffic and vegetation variables. A study in Taiwan, including long-term measurements at 50 locations, showed a statistically significant relationship between land-use types and noise exposure expressed as 24 h average A-weighted equivalent noise levels (L_Aeq,24h_) [23]. Furthermore, a land-use regression (LUR) was developed, which included different variables. For different land-use types and traffic, these LUR models can be used to optimize urban planning taking into account noise pollution as public health problem [24].

In summary, in many countries, the permissibility of specific noise levels is closely connected to land-use categories [19]. These categories are basically defined as recreation zones, zones for exclusively residential use, mixed residential use, and zones without residential use reserved for noisy activities. Planning values, guiding values, or limit values are often graduated in 5 dB steps, which allows in mixed residential areas 5 dB higher noise exposure than in exclusive residential areas. This setting is not supported by studies and is caused by tradition. It remains unclear if these attributes are linked with a different noise rating of the inhabitants? The difference is explained by a lower expectation of noise [25,26] or the higher ambient noise level in mixed-use areas [27].

The Federal Commission for Noise Abatement in Switzerland recently published proposals for the amendment of the Swiss Noise Abatement Ordinance [28]. According to these proposals, purely residential areas and mixed residential areas should be classified with the same noise sensitivity level. This is intended to guarantee the affected residents the same degree of protection. The question arises whether this approach is not only justified from a pragmatic point of view but is also based on the fact that neither noise sensitivity nor noise reaction differs in a relevant way. This research question was investigated by the use of existing data sets from two recent surveys in the state of Tyrol [29,30].

The primary aim of this study was to identify differences between the exposure–response curves (ERC) for pure residential versus mixed-use land category. To understand possible reasons, the self-reported sensitivity and noise annoyance for different sources (traffic, industrial, neighborhood, and total noise) was compared for both categories.

## 2. Materials and Methods

### 2.1. Data Acquisition

Data from two surveys performed during the last years in the state of Tyrol [29,30] were used. These studies included both rural and urban environments. The primary aim of the study in the Lower Inn Valley was the evaluation of the new 4-track railway, where two new tracks were mainly constructed subsurface. Data were already used to investigate the combined effect of road and rail traffic noise exposure on total noise annoyance [12]. The Investigation in Innsbruck [29], the capital of the state of Tyrol, gained data for a total noise investigation, where the elaborated annoyance equivalence model led to evidence-based results [11]. The survey data from Innsbruck are grouped as “urban”, and the dataset from Lower Inn Valley as “rural”. Parameters used in both surveys and potential covariates were included in the whole dataset. These are gender, age, self-reported sensitivity to noise, self-reported state of health, and the highest level of education.

Both studies had similar methods and accuracy in gaining exposure data. The exposure was expressed according to the environmental noise directive (END) [31] as L_den_ and L_night_. The exposure data is based on prediction with the national standards [32,33] which are compatible with the use according to the END. Receiver points were set on the most exposed façade at a height of 4 m above ground. The respondents of the face-to-face interview were linked to these façade levels.

In both studies, sampling took care of demographic balance, spatial distribution, and noise exposure classified into low, medium, and high. While in the Innsbruck study all three noise sources road, rail, and air traffic were considered in the exposure clusters, in the Lower Inn Valley only the impact of rail traffic noise was used. The inclusion criteria for participation in the survey were a primary residence for at least one year and an age of at least 18 years.

In Innsbruck, in the year 2017, the number of respondents was 1031 and a response rate of 48% was achieved. In the Lower Inn Valley where the surveys were conducted from November 2015 to January 2016, there were 1003 respondents and a response rate of 56%.

The formulation of questions about exposure and annoyance followed the recommendations of the ICBEN (International Commission on Biological Effects of Noise). An 11-point scale was used, also for the subjective assessment of personal sensitivity to noise. The Office of the Tyrolean Regional Government, department für spatial planning and statistics, matched the georeferenced receiver points with the officially specified land-use categories. Two groups of land-use categories were defined. The first group contains the category for residential use only. It is characterized by the absence of any commercial activities with extensive sound emissions. The second group summarized all mixed uses, where residential use is combined with activities of craft, commerce, trade, agricultural or touristic use.

Only variables and covariates used in both studies were included. The independent variable was exposure data as L_den_, with no data available for the rural setting for aircraft noise. The dependent variable was the response to noise annoyance by source and in total on a scale of 1 to 10. Age, gender, self-reported noise sensitivity, self-rated health, and highest education level were entered as covariates. Information on socioeconomic status was collected in neither study and therefore not available.

### 2.2. Statistical Analyses

The questionnaire parameters were tested to see if there are significant differences by land-use categories. After checking the distribution function using the Shapiro–Wilk test, this was performed using the Mann–Whitney U test. The selection of covariates for the exposure–response curves (ERC), was calculated in a saturated multivariate binary logistic model using the Wald test [34]. Here, the influences of noise exposure and covariates on the dichotomized annoyance response applicable to the noise source for highly annoyed were analyzed. All answers on the 11-part scale with 8, 9, and 10 were interpreted as highly annoyed, and the other answers as not highly annoyed.

The ERC was a result of multivariate analysis. In addition to the classic epidemiological parameters of age and gender, only the self-reported sensitivity to noise was included in this analysis.

The exposure–response curves were calculated for L_den_ according to the three sources of traffic noise. In a generalized linear model (GLM), source-specific exposure–response curves for the proportion of highly annoyed persons were calculated using binary logistic regression analysis. This was performed separately for the different land-use categories.

## 3. Results

An overview of the descriptive statistics and the main parameters of the available data sets are summarized in Table 1. The distribution between persons living in urban or rural environments in total is almost equal, especially in the mixed land-use category is with 62% versus 38% slightly more linked to urban settings. Differences in the distributions are small for gender, noise sensitivity, and health status. A moderate difference is visible for the age distributions, as the highest proportion of persons in residential areas are in the mid-age group (41–60 years) while for mixed land use the younger persons group shows the highest amount (18–40 years).

The distribution of noise exposure by source and land-use category is shown in Figure 1. For road noise, the majority of persons in residential zones are exposed to L_den_ values between 50 and 60 dB, while in the mixed land-use category the exposure is in general higher, with the highest proportion within the 61 to 65 dB bin. Rail exposure is lower in both land-use zones. The number of persons exposed to L_den_ values higher than 60 dB is almost negligible. Air traffic noise exposure was only evaluated for the urban setting which is the agglomeration of Innsbruck with its airport. Therefore, the number of exposed persons to aircraft noise is lower in Figure 1c. Aircraft noise exposure is in general lower compared to road noise. Again, there is a shift of the distributions between the residential and mixed categories.

Univariate comparisons with the land-use category show a significantly higher annoyance by all sources and explicitly for road traffic and commerce and industry noise for persons living in a mixed area. Self-reported sensitivity to noise, health status, and education is not correlated with living in a certain land-use category as shown in Table 2.

The test results shown in Table 2 refer to the 11-point metric scale for the annoyance responses. There is a significant correlation with the land-use category for annoyance by all sources, by road traffic, and by commerce and industry, while all other items are not significant.

Table 3 shows these comparisons with urban and rural settings. Annoyance by road, rail traffic, and commerce and industry is higher for persons living in rural conditions, while air traffic noise and neighbors’ noise are more dominating annoyances in urban areas.

In Table 4, the results of the Wald test show that, with respect to the corresponding exposure, exclusively noise sensitivities contribute significantly to annoyance for all three traffic noise types. Neither ‘highest education level’ and ‘subjective estimation of health status’ nor the interaction term for the land-use category and for rural versus urban contributes significantly.

After the exclusion of the covariates ‘highest education level’ and ‘subjective estimation of health status’ the final multivariate model contains the corresponding exposure, age, gender sensitivity to noise and setting for the two land-use categories. The setting was not included in the analyses of air traffic noise because of missing exposure data in the rural data.

Multivariable analyzes of self-reported noise sensitivity showed neither significant contribution of exposure levels nor land-use categories and rural or urban settings and their interaction terms.

Figure 2 shows the source-specific exposure–response relationship for noise annoyance for “highly annoyed” caused by road, rail and air traffic noise by land-use categories. The covariates age, gender, sensitivity to noise and setting are centered on the mean.

Figure 2 presents the exposure–response curves for the three traffic sources. The findings of Table 2 and Table 3 are again demonstrated. The land-use category has no impact on exposure–response to railroad and aircraft noise. The curves for aircraft noise appear nearly identical, and the curves for railroad noise with overlapping confidence intervals. For road noise the situation is different. For the entire L_den_ range from 40 to 60 dB, the percentage of highly annoyed is lower for individuals living in residential zones, which fits the rank test results of Table 2. The horizontal shift of the curves around L_den_ of 50 dB is almost 5 dB and becomes smaller for L_den_ > 55 dB. Above 60 dB the curve for the mixed land-use zone intersects with the residential zone curve.

## 4. Discussion

The available dataset consists of two large cohorts of persons living in rural and urban areas. In both areas, persons can either live within pure residential or mixed zones.

In a first step, a potential difference between self-reported sensitivity to noise according to living in a dedicated land-use category was tested. The non-parametric Mann–Whitney-U test showed no evidence for that. The hypothesis that noise-sensitive persons try to live in a property that is located in residential-only areas is either false or circumstances do not allow a high amount of noise-sensitive persons to choose an apartment or house in pure living areas. Traffic noise sources can also change over time with increasing traffic and new build rails or railroads.

In the mixed-use areas, noise annoyance is significantly higher across all sources. This is also true for noise annoyance from road traffic and commercial and industrial activities. For both sources mentioned here, this can be explained by the higher noise exposure from these sources associated with mixed-use activities. No such distinction exists for rail and air traffic. Annoyance by noise from neighbors also shows no dependence on the land-use category.

It is interesting to compare the noise annoyance from all sources between the urban and rural settings on the one hand and the pure and mixed residential areas on the other hand. The comparison between pure and mixed residential areas shows a significant difference (see Table 2). This can be explained by the additional noise exposure from trade and commerce. In contrast, the annoyance from all sources in the urban and rural settings shows no statistically significant difference. Here, the stronger annoyance from road and rail in rural areas is compensated by neighborhood noise and aircraft noise. Since self-reported noise sensitivity is the same in both urban and rural settings, with 15% highly sensitive (see Table 1), this strong moderator fails as an explanatory variable. Possibly, the same potential to declare oneself highly annoyed exists for the same noise sensitivity and only a shift within the source mix takes place. However, this cannot be explained with the available data and requires further research.

A good demonstration of all findings Is visible in the graphical exposure–response curves in Figure 2. While curves for railroad and aircraft noise are identical, irrespective of land-use category, the ERCs differ for road traffic noise. The effect is clearly visible up to an L_den_ of 60 dB. Although taking into account the confidence intervals of both curves, the difference is limited to a certain range and quantity.

A limitation of our study is the lack of information about the type of building, façade insulation, floor level, access to quiet façade and other receiver-specific characteristics. This is likely to result in uncertainties regarding the annoyance response in different settings.

In the case of rail traffic noise, the setting and circumstances have a significant influence. This can be explained by the fact that the survey in the Lower Inn Valley focused on a rail transport project. Therefore, an overestimation of rail traffic noise can be assumed.

The lack of data on socioeconomic status is also a limitation of this study. It should be noted, however, that in the multivariate analysis there was no significant correlation between educational level and annoyance [35]. Since educational level and socioeconomic status are strongly correlated, it can be assumed that the bias due to the lack of information on SES is limited.

Interviewees were selected according to specific criteria based on noise exposure, age, community affiliation, and gender. Compared to online surveys, these characteristics can be better controlled this way. In telephone surveys, it is difficult to reach certain groups of people because their telephone numbers are not public. This also causes a bias in the selection of interviewees.

The strengths of the two underlying studies are the accurate determination of noise exposure. The noise position was selected according to the state of the art for each receiver point and then the selection of the interview cluster was performed. Errors due to misclassification are thus minimized. The used national sound prediction method takes into account the individual emission based on traffic and road/track data, ground and shielding effects, reflections, and atmospheric attenuation. They are state-of-the-art and compatible with interim European prediction methods prior to the just recently introduced common noise assessment methods in the EU (CNOSSOS-EU). Cluster selection within the entire investigated area is only feasible by calculating noise levels for each individual building. Calculations allow the separation of individual sources while with measurements, the separation of individual sources and background noise is time-consuming and often impossible. Measurements are usually also restricted to a comparatively short time section and do not allow assessing a noise index for all day periods of a year, a representative yearly traffic average, and defined meteorological conditions. In summary, the calculation of the noise index L_den_ allows the comparison to the recent EU regulation and the majority of the literature.

Another strength of both studies is the use of face-to-face interviews as the survey method. This approach minimizes selection bias. The exact spatial assignment of the survey data to the land-use categories via point-to-point relationships ensures proper identification.

However, various uncertainties can have an impact on the results of such exposure–response curves, which has been demonstrated in a previous theoretical analysis [36]. A more detailed in-depth analysis regarding uncertainties has been published recently by Horonjeff [37]. While the slope of ERCs changes with increasing uncertainties in noise level determination, also the range of the underlying available exposure data and their distribution within the different sound level bins is of importance. However, the different distributions of exposure levels between residential and mixed land-use categories in context with the findings of Horonjeff could not explain the difference of the observed ERC by uncertainties. The distribution of respondents between those very sensitive to noise and those not sensitive to noise is exactly the same in the residential use and mixed-use categories, 15:85. Self-reported noise sensitivity is the only relevant variable for adjusting the ERC, which was available in both datasets. This is shown by the result of the multivariate logistic regression in Table 4. It acts as a covariate according to Kroesen et al. [14] and not as a confounder, since noise sensitivity is not associated with exposure. Due to the equal distribution in both categories, there is no uncertainty contribution in the comparison of both curves. Therefore, it can be assumed that the ERC results in general are influenced by uncertainties, but the relative positions between the residential and mixed curves remain valid.

A potential weakness of this analysis lies in the different survey periods of the two studies used. While in Innsbruck the surveys took place in 2017, this was conducted in the Lower Inn Valley in 2015/2016. However, there are no reasons to consider substantial changes in traffic events and the associated noise exposure in the years of observation, especially as they did not fall during the period of the Corona pandemic. Due to the comparable average traffic flows, no relevant influences on the exposure side are to be expected. The survey period within the year could have a greater influence on the effect side. While in the Lower Inn Valley, the surveys took place between November and January, in Innsbruck they were conducted from May to June. During spring when people are already more outside and keep their windows more open, people could tend to report more annoyance. This could lead to an underestimation of traffic noise annoyance in the setting of the Lower Inn Valley. However, this possible underestimation effect cannot be seen in the comparison of ERC between Innsbruck and Lower Inn Valley [12], as the response behavior may have been overlaid by the project in Lower Inn Valley that was ongoing during the survey.

Defining traffic noise limits without a distinction between explicit and mixed residential areas as recommended by the Federal Commission for Noise Abatement in Switzerland is supported by the equal self-reported sensitivity to noise and comparison of ERCs of this study.

Such a distinction from the legal point of view can only be argued based on other considerations such as economic reasons and the protection of silent areas from new build infrastructure. For example, in residential zones, noise levels should not be increased to avoid shifts in property prices or living conditions in general, but it is not valid to base these arguments on the noise sensitivity to traffic noise of persons living already there. It is also important that persons living in mixed areas, with already higher noise levels, need the same attention in regard to noise policies and protection measures, as those in quieter residential areas.

## 5. Conclusions

Based on these studies there is no evidence in the exposure–response relationships for traffic noise between the land-use categories residential only and mixed-use. It seems not appropriate to distinguish between both categories when defining limit values to protect neighbors from traffic noise. While it is not valid to allow higher limits for persons living in mixed areas there is no reason to invest in more protective measures just for residential areas. Both land-use categories need the same attention in regard to noise policy and protection measures. Different regulations are possible but must be based on different arguments such as noise response or noise sensitivity.

## Figures and Tables

**Figure 1 ijerph-19-15444-f001:**
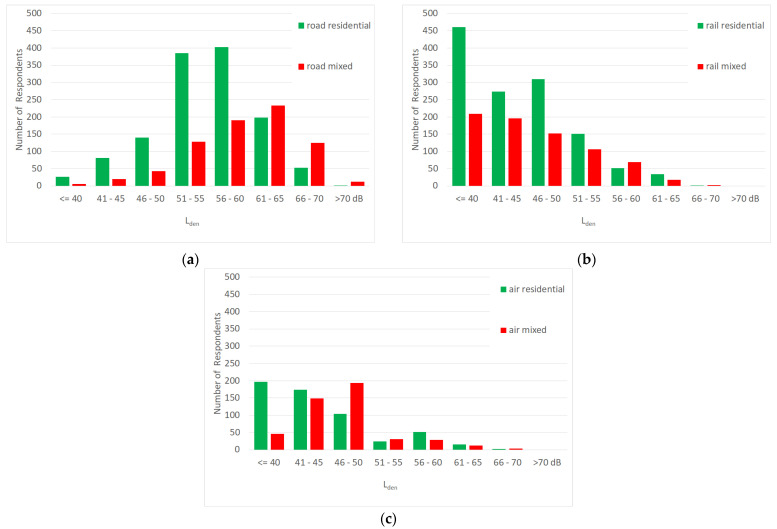
Distribution of noise exposure by land-use categories solely residential (green) and mixed-use (red) for (**a**) road traffic noise; (**b**) rail traffic noise; and (**c**) air traffic noise.

**Figure 2 ijerph-19-15444-f002:**
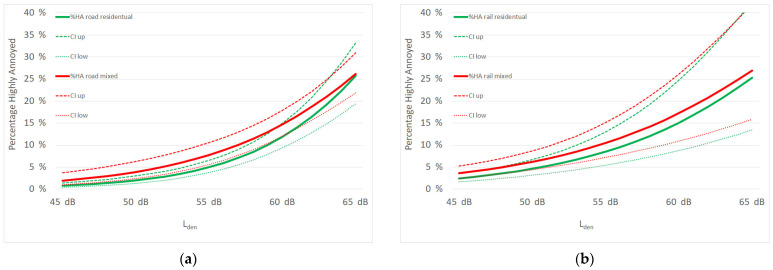
Exposure–response curves by land-use categories solely residential (green) and mixed-use (red) for (**a**) road traffic noise; (**b**) rail traffic noise and (**c**) air traffic noise.

**Table 1 ijerph-19-15444-t001:** Overview of the descriptive statistics for the main parameters of the surveys.

		Residential		Mixed		Total	
		Count	Frequency	Count	Frequency	Count	Frequency
Setting	Urban	568	44%	463	62%	1031	51%
	Rural	714	56%	289	38%	1003	49%
%HA ^(1)^ all sources	Urban	50	4%	75	10%	125	6%
	Rural	81	6%	46	6%	127	6%
%HA road	Urban	32	2%	85	11%	117	6%
	Rural	91	7%	52	7%	143	7%
%HA rail	Urban	14	1%	16	2%	30	1%
	Rural	46	4%	26	3%	72	4%
%HA air	Urban	62	5%	59	8%	121	6%
	Rural	22	2%	9	1%	22	2%
%HA commerce and ind.	Urban	4	0%	11	1%	15	1%
	Rural	21	2%	12	2%	33	2%
%HA neighbors	Urban	22	2%	30	4%	52	3%
	Rural	18	1%	8	1%	26	1%
Gender	Female	670	52%	375	50%	1045	51%
	Male	612	48%	377	50%	989	49%
Age group	18 to 40 years	433	34%	312	41%	745	37%
	41 to 60 years	517	40%	240	32%	757	37%
	over 60 years	332	26%	200	27%	532	26%
Sensitivity	Not highly sensitive	1091	85%	636	85%	1727	85%
	Highly sensitive	191	15%	116	15%	307	15%
Health status ^(2)^	Not good	324	25%	185	25%	509	25%
	Good	958	75%	567	75%	1525	75%
Education ^(3)^	Lower education	687	54%	402	53%	1089	54%
	Higher education	595	46%	350	47%	945	46%
	Total	1282	100%	752	100%	2034	100%

^(1)^ Highly sensitive scores 8 to 10 on the 11-point scale; ^(2)^ Health status “good” scores “good” and “very good” on the 5-point scale; ^(3)^ Education scores high school and university for “higher education” on the 5-point scale.

**Table 2 ijerph-19-15444-t002:** Questionnaire parameters by land-use category—results of the non-parametric Mann–Whitney-U test.

	Overall		Mean Rank	
	Z	*p*	Residential	Mixed
Annoyance by all sources	−6.142	**0.000**	956.66	1121.22
Annoyance by road traffic	−6.888	**0.000**	949.46	1133.50
Annoyance by rail traffic	−1.184	0.237	1028.22	999.22
Annoyance by air traffic	−0.713	0.476	510.09	523.25
Annoyance by commerce and industry	−4.127	**0.000**	987.17	1069.21
Annoyance by neighbors	−1.410	0.158	1004.48	1039.69
Self-reported sensitivity to noise	−0.541	0.588	1022.35	1007.86
Self-reported health status	−1.004	0.316	1025.93	1000.46
Highest education	−0.295	0.768	1017.37	1009.61

Note: *p* asymptotic significance 2-tailed; values lower than the defined significance level of 0.05 are highlighted in bold.

**Table 3 ijerph-19-15444-t003:** Questionnaire parameters by urban or rural setting—results of the non-parametric Mann–Whitney-U test.

	Overall		Mean Rank	
	Z	*p*	Urban	Rural
Annoyance by all sources	−1.861	0.063	1041.24	993.10
Annoyance by road traffic	−3.716	**0.000**	970.22	1066.10
Annoyance by rail traffic	−11.676	**0.000**	881.25	1157.55
Annoyance by air traffic	−15.782	**0.000**	1213.20	816.33
Annoyance by commerce and industry	−5.779	**0.000**	976.59	1059.55
Annoyance by neighbors	−2.915	**0.004**	1086.20	946.89
Self-reported sensitivity to noise	−1.401	0.161	1054.14	987.78
Self-reported health status	−14.422	**0.000**	1033.45	999.12
Highest education	−1.861	0.063	1194.93	829.02

Note: *p* asymptotic significance 2-tailed; values lower than the defined significance level of 0.05 are highlighted in bold.

**Table 4 ijerph-19-15444-t004:** Results of the logistic regression for highly annoyed caused by specific noise exposure and covariates.

	Regression	Standard	Wald	*p*	OR	OR	
	Coefficient B	Error				CI−	CI+
Annoyance by Road Traffic Noise as %HA							
L_den.road_ (score change per 1 dB increase)	0.167	0.015	127.132	**0.000**	1.182	1.148	1.217
gender (male vs. female)	−0.297	0.149	3.981	**0.046**	0.743	0.556	0.995
age (score change by 1 year increase)	0.007	0.005	2.549	0.110	1.007	0.998	1.016
sens (score change by 1 point increase)	0.264	0.028	85.830	**0.000**	1.302	1.231	1.377
edu (lower vs. higher education)	0.046	0.060	0.583	0.445	1.047	0.931	1.178
health (not good vs. good health status)	−0.036	0.083	0.188	0.665	0.965	0.820	1.135
land-use (residential vs. mixed)	0.434	0.248	3.067	0.080	1.543	0.950	2.508
setting (urban vs. rural)	0.438	0.238	3.386	0.066	1.550	0.972	2.472
land-use × setting	−0.445	0.317	1.967	0.161	0.641	0.344	1.193
intercept	−13.398	0.971	190.508	**0.000**	0.000		
Annoyance by Rail Traffic Noise as %HA							
L_den.rail_ (score change per 1 dB increase)	0.123	0.015	67.172	**0.000**	1.130	1.098	1.164
gender (male vs. female)	0.321	0.217	2.178	0.140	1.378	0.900	2.110
age (score change by 1 year increase)	−0.008	0.007	1.209	0.271	0.992	0.979	1.006
sens (score change by 1 point increase)	0.205	0.040	25.741	**0.000**	1.228	1.134	1.329
edu (lower vs. higher education)	0.092	0.091	1.005	0.316	1.096	0.916	1.311
health (not good vs. good health status)	0.222	0.119	3.462	0.063	1.249	0.988	1.578
land-use (residential vs. mixed)	0.204	0.386	0.281	0.596	1.227	0.576	2.611
setting (urban vs. rural)	1.412	0.340	17.253	**0.000**	4.106	2.108	7.995
land-use × setting	−0.066	0.476	0.019	0.890	0.936	0.368	2.379
intercept	−11.535	1.073	115.560	**0.000**	0.000		
Annoyance by Air Traffic Noise as %HA							
L_den.air_ (score change per 1 dB increase)	0.130	0.015	73.597	**0.000**	1.139	1.106	1.173
gender (male vs. female)	0.290	0.211	1.878	0.171	1.336	0.883	2.022
age (score change by 1 year increase)	0.010	0.006	3.037	0.081	1.010	0.999	1.022
sens (score change by 1 point increase)	0.217	0.040	29.068	**0.000**	1.243	1.148	1.345
edu (lower vs. higher education)	0.082	0.085	0.926	0.336	1.086	0.918	1.283
health (not good vs. good health status)	0.211	0.109	3.774	0.052	1.235	0.998	1.529
land-use (residential vs. mixed)	0.024	0.211	0.013	0.911	1.024	0.677	1.548
intercept	−10.905	1.043	109.369	**0.000**	0.000		

Note: Degree of freedom = 1; *p* = significance; OR = Odds Ratio; CI− = lower 95% confidence interval; CI+ = upper 95% confidence interval; *p*-values lower than 0.05 are highlighted in bold; sensitivity scores on 11-point scale.

## Data Availability

Not applicable.

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
