# Peer review of "The Effect of Land-Use Categories on Traffic Noise Annoyance"

_ijerph, 2022, doi:10.3390/ijerph192315444_

Round 1
Reviewer 1 Report
This study addresses the impact of different land use categories on the annoyance response to environmental noise. The rationale behind this study is the use of land use specific noise limit values in some countries, and the research question is whether this practice is justified by knowledge regarding factors that affect noise annoyance beyond the level of noise exposure. The study is based on two noise surveys in urban and rural areas of Austria (Tyrol) subdivided between pure residential areas and mixed areas (residential + commercial).
The research questions are very interesting both with respect to the need of more knowledge about contextual effects on noise annoyance responses as well as regarding implications for noise policy. Strengths of the study are the applied focus and policy relevance and the use of two surveys linking noise responses to modelled noise levels for the main sources of environmental noise – road traffic, rail and aircraft noise. There are however several limitations in the analytical approach, statistical analyses and the interpretation of the results that need to be addressed. First, there is a lack of coherence between focus of the introduction and the research questions. Second, the statistical approach is weak and unmotivated and, in my opinion, too many univariate results are reported. Lastly, the discussion lacks a comparison with previous findings as well as a proper discussion of the possible limitations of the study and the generalizability.
Specific comments:
Abstract
- Please moderate to: Land use categories are sometimes/often used to define exposure limits…
- Please replace the term “threshold levels” to “noise limit values” or “guideline values for noise” since a threshold is the level at which there is an observed measured effect.
Introduction
- The introduction is poorly written and fragmentary. The introduction should include a motivated and up-to-date summary of the knowledge about which factors affecting noise annoyance in general and more specifically about what the scientific literature shows regarding different types of built environment factors including land use.
- Individual studies are apparently arbitrarily referred to without it being clearly stated what are the relevant research findings that form the background for this study, e.g., the reference to the study conducted in Halifax seems unmotivated (line 37). The study is just mentioned with no reporting of findings of relevance for the present study. The study from Canada did, as far as I could see, not measure annoyance, only noise levels in different residential areas. Furthermore, it is referred to Italian legislation. Where is this legislation adopted to? Further, you refer to the SIRENE study that showed that access to greenness impacted on annoyance. Please define “green metrics” as this is not self-explaining. Apart from the general findings that built environment factors such as green space impact on noise annoyance, why is this particular result relevant for your study. Green space is not included in your study, neither included in your discussion as a possible limitation that you did not had access to this information in your study).
The introduction should also include more information regarding previous studies reported ER curves for noise and annoyance e.g. Guski, R., et al. 2017. "WHO environmental noise guidelines for the European region: A systematic review on environmental noise and annoyance." International Journal of Environmental Research and Public Health 14(12). It should also be noted that ER curves for noise annoyance are known to differ between areas and that several non-acoustic factors impact on these associations.
- You state that it is some evidence that noise may be differently distributed across communities based on socio-economic position (SEP), but no further information or results, e.g., with regards to annoyance responses, are given. Furthermore, that LUR models are used to model noise exposure. I cannot see why these studies in particular are relevant for your study. LUR models are seldom used to calculate noise from transportation, at least in Europe where source specific noise exposure assessment is predominant. Thus, this model and the results from this study does not seem particularly relevant for Europe and European noise legislation as well as for the scientific basis.
- Please consider skipping the listing of all noise indicators that are used in different countries as this is not so interesting for your research context. Try instead to summarize or to state that the noise legislation differs across the European countries, both with regard to noise indicators (Lden, Lnight, LAmax etc.), the level of the noise limits as well as that in some countries land use categories are used to differentiate the noise limits.
- You should also refer to the recent WHO guidelines for environmental noise which gives source specific recommendations for Lden and Lnight based on systematic review on noise and health, including noise annoyance. These recommendations are not differentiated based on land use, but similar for all kinds of residential areas. The systematic review on noise and annoyance by Guski et al., 2017 formed the basis for the WHO guidelines.
- Please also introduce the urban rural perspective as this is analyzed in your study. What does the literature tell us about that?
- Please introduce noise sensitivity; what is it and how does it impact on the annoyance response. The paper by van Kamp., et al. (2004). "The role of noise sensitivity in the noise-response relation: a comparison of three international airport studies." J Acoust Soc Am 116(6): 3471-3479, is highly recommended as an introduction to noise sensitivity and how it is related to noise annoyance and noise exposure.
- Some factors that have shown to impact on the noise exposure levels, such as SES, vegetation are mentioned and specific papers are referred to. Since your main research question/aim is to investigate whether ERC differ between populations in different types of residential areas, I recommend focusing on previous literature on which factors that are found to impact on noise annoyance and not only on the noise exposure.
- Line 87: It is questionable that the research question should be investigated by use of existing data from surveys from Tyrol. Please reformulate! Annoyance due to noise from a specific source has shown to differ between areas, due to unknown, non-acoustical contextual factors. You have already collected data, but these factors could be properly investigated also in several other areas. The studies by Lercher et al noise annoyance in alpine Walley??
- Please reconsider your research questions and base it on a thorough literature review. It is research question 4 that is the interesting one with respect to the overall aim of the paper and with respect to noise policy. Thus, this needs to be addressed more properly. Do the exposure -response curves (ERC) differ significantly between different types of land use categories? The other research questions are related but could be more properly treated in a multivariate model (see comments below on Statistical analyses)
Materials and methods
Data acquisition
- Please include information about the method of recruitment, eligible study population and response rate.
- Please include information about the noise annoyance question (standard questions?) as well as the noise sensitivity question(s)?
- Please also give more information about how the variables were categorized, including the covariates.
- Among your included covariates, which should be based on a thorough literature review (see my comments above), did you have access to household income? If yes, why was income not considered? Some studies have reported income to be associated with noise exposure.
Statistical analyses.
- You report to have done univariate correlations, but you have not performed correlation analyses but tested if the median (e.g. annoyance) is different between two groups. Please be correct when describing your analyses.
- The main aim and interest of this paper is the exposure-response associations. I suggest you skip the univariate analyses and focus on the multivariate model (Aim 4 above), as the initial univariate analyses add little to the main results. You can still give descriptive results in tables, but it is more informative to the reader that you report on e.g. % HA, for different groups (land use and urban versus rural) and there is no need for a statistical significance test.
- Instead of all the statistical test, consider developing a directed acyclic graph (DAG) for selection of your covariates. A DAG should be based on prior knowledge or a priori assumptions about causal relations among variables of interest in your study population. A thorough review of which factors that impact on both noise exposure and noise annoyance will also be a good basis for improvement of the introduction. Based on your DAG include the relevant covariates into a multivariate regression model (logistic regression model) and run the analyses separately for air, road and rail. The analyses could then be stratified for land use and for rural versus urban. If assumed that these variables could be effect modifiers, you could include them as interaction terms in your regression model.
Results
- You could also include %HA in your Table 1 as this table was supposed to show descriptive statistics for all your main parameters. Consider skipping the univariate test and results (Table 2 and 3), as they add nothing to the main results, and it is not correlations either, it is a mere test of differences between two populations.
- What is the purpose of the analyses and results shown in table 4? As far as I can see it only shows that noise exposure is associated with annoyance and that noise sensitivity is associated with annoyance. These are well known associations, and as long as they are univariately tested they do not add so much to the interpretation (see also my comments to the statistical approach).
- Table 5 shows your main results. It is not clear whether the results are based on a multivariate model including all the covariates and that each effect parameter is adjusted for all the covariates. The effect parameter exp (B) is more commonly termed odds ratio (OR). Another more common method in epidemiology is to report the the crude effect estimate (OR) without adjusting for covariates (or only age and sex) and then include more covariates in successive models and observe whether the OR is changing; depending on your previous set of covariates.
- Please include the ER functions for your ER curves, and the fitting model.
Discussion
- A limitation of your study is that you have no information about type of building, façade insulation properties, access to a quite façade etc. that may be different between your study areas and that most certainly will impact on the actual noise exposure and hence the annoyance responses. This needs to be addressed. You state that your noise model is accurate, but there will always be uncertainty in the noise exposure assessment, that needs to be mentioned.
- Please elaborate on how face to face interview can reduce selection bias? This needs to be discussed in light of your response rate and whether you assume that your study sample is representative for your study areas.
- Please use exposure-response curve (not exposure effect curve) to be correct and consistent throughout the paper.
- Your interpretation of the results are based on speculative assumptions and lack references to previous reported results, e.g. you state that noise sensitivity is the only relevant variable to adjust for, but you have not tested this in a multivariate model, wheter it impacts on your effect estimate. Previous studies have shown that noise sensitivity is associated with noise annoyance, but not with the noise exposure, thus it is not a confounder in the association between noise exposure and annoyance. More about the role of noise sensitivity in this association can be read in the paper of van Kamp et al., 2004.
- There are too much discussions of your univariate results which could easily be biased. Please focus the discussion on your main analytical results and the ER curves.
- I miss a comparison of your ER curves with previously reported ER curves for HA, eg. those reported by Guski et al., 2017 from meta-analyses to provide knowledge for the WHO environmental noise guidelines, but also other studies from Austria, reported by Lercher, P. et al. (2008). A comparison of regional noise-annoyance-curves in alpine areas with the European standard curves. In Proceedings of the 9th International Conference on Noise as a Public Health Problem (ICBEN 2008), Foxwoods, CT, USA, 21–25 July 2008. Results from these study areas are also included in the meta-anlyses by Guski et al., 2017.
- You state that a weakness could be different survey periods for the two studies. Could different seasons for obtaining the survey responses have impacted on your results? During summer when people are more outside and keep window more open, people could tend to report more annoyance. Please discuss.
Author Response
Thank you for your valuable comments, which helped to substantially improve the manuscript. Please see our responses in the attachment.
Kind Regards

Reviewer 2 Report
The paper is quite interesting and well written. Research on this topic is very important and necessary. Such research has already been carried out in the world, but this paper brings new knowledge of the effect of two types of land use categories (residential and mixed uses) on traffic noise annoyance (3 types of noise sources were distinguished).
The structure and layout of the paper and its completeness - without reservations. Research methodology and analysis of the results - correct. The statistical analysis is at a very good level. The editorial and graphic side of the work do not raise any major objections. However, I suggest a few editorial changes:
# In table 1, last row - the percentages should be corrected (63% and 37% replaced by 100%), as the percentage refers to the type of the area in the column.
# In Figures 1 and 2, the descriptions of the horizontal and vertical axes should be entered. Also pay attention to "cutting" the legend.
Author Response
# In table 1, last row - the percentages should be corrected (63% and 37% replaced by 100%), as the percentage refers to the type of the area in the column.
# In Figures 1 and 2, the descriptions of the horizontal and vertical axes should be entered. Also pay attention to "cutting" the legend.
Dear Reviewer,
Thank you for the appreciative review.
Both suggestions were improved in the manuscript. The axes have been labeled and the legends are no longer cut by the curves.
Kind regards
Reviewer 3 Report
the article describes the study of two zones previously defined according to their category of use
some recommendations to be modified :
-Line 38. bibliographical reference about Italian legislation
-Line 47-48 "quiet green space seem to be more effective in rural area" -- Any study or project that confirms this assertion?
- Line 99: Eliminate "we" , please use impersonal person
-Line 145. Please the title appears on the same page as the table
- Table 1. total results: the % values eliminate ".0"
- Title of figure 1 place it on top of the charts
- Table 3 appears complete on the same page
-Line 192-193 rewrite , similarly to line 184-187
Table 5 appears complete on the same page or or at least the title "annoyance by air traffic noise" is on the same sheet
Line 220 don´t -> do not
Line 241, the text mentions response curves, which curves does it refer to?
Line 265 -> "we" use impersonal
Author Response
Thank you for your valuable comments, which helped to improve the manuscript. Please see our responses in the attachment.
Kind regards

Reviewer 4 Report
1. Abstract, Lines 13-14: The definition of mix use is unclear and has to be modified.
2. Abstract, Line 23: Please correct the term “defined” as “define”.
3. Introduction, Lines 28-30: Please provide the related reference.
4. Page 1, Lines 43-45: The sentence is not clear and should be rewritten.
5. Page 2, Lines 67-70: Some studies related to land use type or land use regression with environmental noise measurements were missed in the literature review and could be cited (i.e., Wang VS et al, 2016. Environmental Pollution, 219:174-181; Chang TY et al, 2019. Environment International, 131:104959).
6. Page 2, Lines 89-96: These sentences should be rewritten as one paragraph to clarify the purposes of this study.
7. Page 3, Lines 100-102: This sentence is not clear and has to be clarified.
8. Page 3, Lines 116-118: There are grammatical errors in this sentence and should be corrected.
9. Statistical analysis: Please explain the use of the Kolmogorov-Smirnov test rather than Shapiro-Wilk test to evaluate the normality of data. How to decide which parameters (i.e. age, gender and the self-reported sensitivity) should be considered in the analyses? In addition, the social economic status (SES) is also related to the noise annoyance. The authors should take this factor into account in the model.
10. Table 1: The authors should show the significant but not only qualitative results between these two land use categories for various demographic variables.
11. Figure 1: The statistical differences in road, rail, and air traffic noise levels between residential and mixed land use categories are also needed to present in the figure.
12. Table 5: The odds ratio (OR) and its 95% confidence interval (CI) should be present to replace the current format of Ex(B) and lower and upper values.
13. Discussion, Lines 218-221: This sentence is too long to be understood. It should be modified.
14. Discussion, Lines 244-245: A “comma” should be added in this sentence.
15. Discussion, Lines 246-247: This study did not provide the accurate determination of noise exposure because the exposure data were depended on prediction with the national standards instead of field measurements or monitoring data from noise stations. It is hard to agree that this point is the strength of this study.
16. Discussion, Lines 272-274: This sentence is unclear and has to be rewritten.
Author Response
Thank you for your valuable comments, which helped to improve the manuscript. Please see our responses in the attachment.
Kind regrards

Round 2
Reviewer 1 Report
Most of my concerns and comments to the paper are adequately addressed by the authors. There are however some issues that still are poorly addressed. I have also some suggestions for corrections.
Line 96: Please change LA,den to Lden. Even though Lden is an A-weighted measure, the official indicator name is Lden.
Line 142: Please reformulate from This should be… to This research question was investigated by use of existing data sets from two recent surveys in the State of Tyrol.
Line 317: The sentence starting with the first question can be skipped/reformulated according to your changes in the Introduction, where these specific questions were deleted.
Line 357: Please correct SOS to SES.
Line 358: “Criteria” is perhaps a better description than “rules”?
Line 392: You still state that “Self-reported noise sensitivity is the only relevant variable for adjusting the ERC”. Why and How? Please explain this in light of the fact that noise sensitivity is not a confounder in a multivariate model.
Line 405: Your note that This could lead to an underestimation of traffic noise in the setting of the Lower Inn Valley needs to be rephrased. This contradict what you state above that there is no reason to consider substantial changes in traffic events within the study period. Thus, it would presumably be more correct to state that the actual noise exposure (due to opened windows and more time spent outdoors), is underestimated and thus possibly the annoyance response will be higher for the urban sample compared to the rural sample. You should further elaborate on how the seasonal differences could possibly impact on the urban -rural differences that you found.
Author Response
Thank you very much for your review. We have adapted the document accordingly. The answers to the individual comments can be seen in the attached document.
Kind regards

Reviewer 4 Report
1. Please provide a citation to support the selection of covariates in the model in the study.
2. The statistical results in comparisons have to be mentioned in the text. For instance, were there significant differences in %HA all sources, %HA road, age group, education etc. between residential and mixed areas? Please also define the HA here.
3. It is not common to show the results as Table 2 and Table 3. For instance, why did the authors want to show the Z values here? What did the “mean rank” mean for urban and rural? Did the authors intend to show the difference in mean rank between urban and rural areas? The presented format of tables should be consulted by a statistician or epidemiologist.
4. Please show all significant differences in different noise groups (i.e., 51-55 dB, 56-60 dB in 1(a); <=40 dB, 46-50 dB in 1(b); <=40 dB, 46-50 dB in 1(c) etc.) between residential and mixed areas by noise sources in Figure 1.
Author Response
Thank you very much for your review. We have amended the document accordingly. The answers to the individual comments can be seen in the attached document.
Kind regards
